# Rescue of Aberrant Splicing Caused by a Novel Complex Deep-intronic *ABCA4* Allele

**DOI:** 10.3390/genes15121503

**Published:** 2024-11-23

**Authors:** Jordi Maggi, Silke Feil, Jiradet Gloggnitzer, Kevin Maggi, James V. M. Hanson, Samuel Koller, Christina Gerth-Kahlert, Wolfgang Berger

**Affiliations:** 1Institute of Medical Molecular Genetics, University of Zurich, 8952 Schlieren, Switzerland; maggi@medmolgen.uzh.ch (J.M.); feil@medmolgen.uzh.ch (S.F.); gloggnitzer@medmolgen.uzh.ch (J.G.); kmaggi@medmolgen.uzh.ch (K.M.); koller@medmolgen.uzh.ch (S.K.); 2Department of Ophthalmology, University Hospital Zurich and University of Zurich, 8091 Zurich, Switzerland; james.hanson@usz.ch (J.V.M.H.); christina.gerth-kahlert@usz.ch (C.G.-K.); 3Zurich Center for Integrative Human Physiology (ZIHP), University of Zurich, 8057 Zurich, Switzerland; 4Neuroscience Center Zurich (ZNZ), University and ETH Zurich, 8057 Zurich, Switzerland

**Keywords:** ABCA4, Stargardt disease, deep-intronic variant, complex allele, pseudoexon, antisense oligonucleotide, rescue, splicing, minigene, retinal organoid

## Abstract

**Background/Objectives:** Stargardt disease (STGD1) is an autosomal recessive disorder caused by pathogenic variants in *ABCA4* that affects the retina and is characterised by progressive central vision loss. The onset of disease manifestations varies from childhood to early adulthood. **Methods:** Whole exome (WES), whole gene, and whole genome sequencing (WGS) were performed for a patient with STGD1. **Results:** WES revealed a heterozygous pathogenic missense variant in *ABCA4*, but no second pathogenic variant was found. *ABCA4* whole-gene sequencing, subsequent WGS, and segregation analysis identified a complex deep-intronic allele (NM_000350.2(ABCA4):c.[1555-5882C>A;1555-5784C>G]) *in trans* to the missense variant. Minigene assays combined with nanopore sequencing were performed to characterise this deep-intronic complex allele in more detail. Surprisingly, the reference minigene revealed the existence of two pseudoexons in intron 11 of the *ABCA4* gene that are included in low-abundance (<1%) transcripts. Both pseudoexons could be confirmed in cDNA derived from wildtype retinal organoids. Despite mild splicing predictions, the variant minigene revealed that the complex deep-intronic allele substantially increased the abundance of transcripts that included the pseudoexon overlapping with the variants. Two antisense oligonucleotides (AONs) were designed to rescue the aberrant splicing events. Both AONs increased the proportion of correctly spliced transcripts, and one of them rescued correct splicing to reference levels. **Conclusions:** Minigene assays combined with nanopore sequencing proved instrumental in identifying low-abundance transcripts including pseudoexons from wildtype *ABCA4* intron 11, one of which was substantially increased by the complex allele.

## 1. Introduction

Stargardt disease (STGD1; MIM #248200) is an autosomal recessive disease affecting the retina. It is characterised by central vision loss and accumulation of yellow-white flecks (lipofuscin) at the retinal pigment epithelium (RPE) level, with a prevalence of 1 in 8000–10,000 individuals [1,2,3,4,5,6]. The characteristic clinical manifestation of STGD1 is progressive central vision loss with onset either in childhood or in early adulthood [2,3,4,5,7]. Early onset is generally associated with worse prognosis [1,3,5,6,8].

STGD1 is caused by biallelic pathogenic variants in the gene *ABCA4*, located on human chromosome 1 [9]. This gene encodes a member of the superfamily of ATP-binding cassette (ABC) transporters involved in retinoid recycling in the visual cycle [9,10]. The gene is expressed mainly in the retina, specifically in photoreceptors (both rods and cones) and RPE [9,10,11,12]. Additionally, *ABCA4* expression has been detected in human skin and hair follicles [13,14,15]. In the retina, the ABCA4 transporter is localised in the rim regions of rod and cone photoreceptors’ outer segments, and in endolysosomal membranes in RPE cells [10,12]. In photoreceptors, ABCA4 plays an important role in the visual cycle by transporting phosphatidylethanolamine (PE) and *N*-retinylidene-PE (*N*-ret-PE) across disc membranes [16]. In RPE cells, the transporter has been proposed to support clearance of *N*-ret-PE derived from the phagocytosis of distal photoreceptor outer segments [12].

The *ABCA4* gene is composed of 50 exons and covers a genomic region spanning 130 kb. The gene is highly polymorphic, with a wide spectrum of pathogenic variants identified, including missense, nonsense, synonymous, splicing, and regulatory variants, but also small indels, gross deletions, gross insertions and duplications, and complex rearrangements [17,18]. The Human Gene Mutation Database (HGMD) Professional (v.2024.2, accessed 1 August 2024) contains 2365 *ABCA4* variants (1930 in coding regions, and 435 in noncoding regions), of which 1917 are annotated as disease-causing and 435 as possibly disease-causing [19]. Similarly, the Leiden Open (source) Variation Database (LOVD) dedicated to *ABCA4* listed 2246 variants on the 31st of December 2020 [20,21]. Of these, 796 were classified as pathogenic, 452 as likely pathogenic, 804 as variants of unknown significance (VUS), 182 as likely benign, and 12 as benign by a panel of experts following the ACMG classification guidelines incorporating *ABCA4*-specific ClinGen recommendations [20,22]. Finally, the ClinVar database (accessed on 1 August 2024) has records for 4084 variants in *ABCA4*; 944 have been submitted as pathogenic, 578 as likely pathogenic, 949 as VUS, 1180 as likely benign, 247 as benign, and 298 have conflicting classifications.

Interpretation of *ABCA4* variants is crucial for molecular diagnostics in patients affected not only by STGD1 but also other forms of inherited retinal diseases (IRDs), as variants in this gene can be associated with clinical phenotypes that differ substantially from the typical findings in STGD1 [18]. Variants in *ABCA4* are often reported as the most common contributors to disease in large mixed IRD cohorts [23,24,25]. Concomitantly, interpretation of these variants can be complicated, as even synonymous variants, deep-intronic variants, and variants with a relatively high allele frequency can be pathogenic [18,20,26,27,28]. Moreover, phasing candidate variants is important as multiple complex alleles have been described [28,29,30,31]. Nonetheless, establishing a molecular diagnosis for patients affected by STGD1 could give them access to gene-specific therapies that are being developed [32].

Genetic testing for IRD patients leads to a molecular diagnosis in 50–75% of patients [23,24,33,34]. A single heterozygous likely pathogenic variant in *ABCA4* is often identified in undiagnosed patients [24]. To improve diagnostic yield, several studies focused on these patients and tried to identify the second pathogenic variant by sequencing intronic regions to identify deep-intronic variants (which are often not covered during first-tier genetic testing), performing structural variant analysis, or looking for variants with increased frequency within STGD1 patients [27,28,29,35,36,37]. Studies focusing on noncanonical splice site or deep-intronic variants often used minigene assays to verify their pathogenicity [37,38], and occasionally rescued variant-induced aberrant splicing events with antisense oligonucleotides (AONs) [28,29,39]. HGMD (v.2024.2, accessed 1 August 2024) lists 94 disease-causing or possibly disease-causing deep-intronic (located more than 10 nucleotides away from the exon boundary) variants in *ABCA4*.

In this study, we describe the aberrant splicing events caused by a novel deep-intronic complex allele in intron 11 of *ABCA4*, which was identified in a STGD1 patient *in trans* to a pathogenic missense variant. A minigene assay revealed the existence of pseudoexons in wildtype *ABCA4* intron 11 and that the complex allele leads to increased inclusion of one of these pseudoexons in the transcripts, despite mild splicing predictions. Furthermore, evidence for the existence of both pseudoexons was gathered from cDNA of wildtype retinal organoids derived from induced pluripotent stem cells (iPSCs). AON splicing modulation treatment allowed for complete rescue of the aberrant splicing events in minigene assays.

## 2. Materials and Methods

### 2.1. Clinical Examinations

The patient was referred to our tertiary centre from a community ophthalmologist due to reduced vision, eccentric fixation, and foveal thinning. His initial visit to our clinic took place at age 10 years. At this visit he received an ophthalmological exam including the following: slit lamp examination of the anterior eye; dilated fundoscopy; spectral domain optical coherence tomography (SD-OCT) and fundus autofluorescence (FAF) with a Spectralis (Heidelberg Engineering GmbH, Heidelberg, Germany); and recording of the full-field and multifocal electroretinogram (ffERG; MF-ERG) with an Espion system (Diagnosys LLC, Lowell, MA, USA).

SD-OCT volume scans consisted of 31 horizontal b-scans of 30º in length (equivalent to 8.8 mm) vertically separated by 243 µm acquired in high resolution mode using Automatic Real-time Tracking averaging (12 averaged images per b-scan). The ffERG was recorded according to contemporary recommendations of the International Society for Clinical Electrophysiology of Vision [40] using Dawson–Trick–Litzkow (DTL) electrodes (Diagnosys LLC, Lowell, MA, USA) positioned at the lower lid margin. The sampling rate was 2 kHz. The MF-ERG was recorded using an achromatic stimulus array of 61 eccentricity-scaled hexagons and DTL electrodes according to contemporary ISCEV recommendations [41]. The frame frequency was 75 Hz, and the first-order kernels were analysed.

Subsequent ophthalmological examinations were performed 1, 2, 3, 4, and 6 years after the initial examination. SD-OCT and FAF imaging was performed at 2, 3, 4, and 6 years; MF-ERG was recorded again at 2, 4, and 6 years.

### 2.2. Genetic Testing

Informed consent was obtained from the parents of the patient. Genomic DNA (gDNA) was extracted from whole blood samples of the index patient and both unaffected parents with the automated Chemagic MSM I system and the Chemagic DNA Blood Kit according to the manufacturer’s specifications (PerkinElmer Chemagen Technologie GmbH, Baesweiler, Germany). Whole-exome sequencing (WES), whole-gene sequencing using long-range PCR (LR-PCR), and whole-genome sequencing (WGS) were performed as previously described [42,43,44]. Briefly, WES was performed for the index patient with an IDT Exome v2 kit (Integrated DNA Technologies, Coralville, IA, USA) on an Illumina NextSeq instrument (Illumina, San Diego, CA, USA) according to the manufacturers’ instructions. LR-PCR for the entire *ABCA4* locus (8 different PCRs) was performed for the index patient and both parents and sequenced on an Illumina MiSeq instrument (Illumina, San Diego, CA, USA), as previously described [43]. WGS was performed for the index patient using the TruSeq Nano DNA PCR-Free kit on a NovaSeq 6000 instrument (Illumina, San Diego, CA, USA).

Segregation analysis from both parents was performed for candidate variants by Sanger sequencing or LR-PCR, as previously described [42,43].

### 2.3. Variants Prioritisation

Sequencing data were analysed as previously described and the pipeline can be accessed on Github (https://github.com/jordimaggi/WGS_analysis_workflow, accessed on 1 August 2024) [44]. Briefly, raw sequencing reads were aligned to the human reference genome hg19 following GATK v4.2.6.1 Best Practices (https://gatk.broadinstitute.org/hc/en-us/articles/360035535932-Germline-short-variant-discovery-SNPs-Indels, accessed on 16 August 2021) with the Burrows–Wheeler Aligner (BWA) v0.7.17 [45,46]. HaplotypeCaller (from the GATK pipeline) and DeepVariant v.1.3.0 [47] were used for variant calling. Only variants present on both call sets were considered. The merged Variant Call Format (VCF) file was annotated by the Nirvana annotator v3.16.1 (https://illumina.github.io/NirvanaDocumentation/, accessed on 6 October 2021).

Only variants within known IRD-associated loci were retained (Appendix A). Variants were prioritised based on gnomAD frequencies [48], ClinVar entries [49], phyloP score [50], CADD v1.6 score [51], spliceAI scores [52], primateAI score [53], revel score [54], sift prediction [55], polyPhen prediction [56], family history, and in-house frequencies [44]. The HGMD and LOVD databases were queried for entries for candidate variants [19,21]. Finally, ACMG classification for candidate variants was assessed with the Franklin platform (https://franklin.genoox.com/clinical-db/home, accessed on 10 August 2024), and manual curation of Segregation Data evidence (Evidence categories PP1/BS4) to reassess variant classification [22].

### 2.4. Minigene Assay

A minigene assay was utilised to functionally test the effect on splicing of a candidate complex allele in intron 11 of *ABCA4*. The minigene construct is based on the previously described pcDNA3.1 backbone with an insert corresponding to the human genomic region encompassing exons 3 to 5 of the gene *RHO* [42,57,58], which was inspired by a similar construct utilised to characterise *ABCA4* variants [38]. The genomic region surrounding the complex allele (6944 bp of *ABCA4* intron 11, chr1:g.94531516–94538459 on hg19) was amplified by PCR from the index patient’s gDNA with Phusion High-Fidelity DNA Polymerase (New England Biolabs, Ipswich, MA, USA). The following primers were used: forward primer TCTGCAATCTCATTCACCCCATAA, and reverse primer TGATTCTCTCACAAACAGGCATTG. The PCR reaction mix contained 1× Phusion HF Buffer, 0.2 mM dNTPs, 0.5 µM of each primer, 10 ng of gDNA, and 0.02 U/µL of Phusion High-Fidelity DNA Polymerase in a 20 µL volume. The PCR was performed on a Veriti thermal cycler (Applied Biosystems, Foster City, CA, USA) with the following conditions: 98 °C for 1 min; 35 cycles of 98 °C for 10 s, 65 °C for 30 s, and 72 °C for 5 min; and 72 °C for 10 min. Electrophoresis on 1% agarose gels was performed to verify PCR products.

The *Pfl*MI and *Eco*NI restriction enzymes were used to excise *RHO* exon 4 and part of the flanking introns. Subsequently, the Takara In-Fusion HD cloning kit (Takara Bio, Kusatsu, Japan) was used according to the manufacturer’s instructions to introduce the region and variant of interest. The primers used for amplification of the region of interest had overhangs complementary to the sticky ends of the digested pcDNA3.1_RHO backbone (forward primer overhang CCCTGGAGGAGCCATGGTCTGG, and reverse primer overhang TCGGAGGTACCTCTCCGAGG). The genotypes of the complex allele were verified in selected clones by Sanger sequencing, as previously described [42].

A plasmid representing the reference sequence over the complex allele region (referred to as the reference minigene) and one representing the variant sequence (referred to as variant minigene) were selected for transfection into HEK293T cells using the Xfect Transfection Reagent (Takara Bio, Kusatsu, Japan) according to the manufacturer’s instructions. Cells were harvested 48 h post-transfection for total RNA isolation with the NucleoSpin RNA Plus kit (Macherey-Nagel, Düren, Germany). Subsequently, cDNA was obtained from total RNA with the SuperScript III First-Strand Synthesis SuperMix kit (Invitrogen, Waltham, MA, USA) according to the manufacturer’s instructions.

Minigene-specific transcripts from cDNA were amplified by PCR with primers binding to exons 3 and 5 of *RHO* cDNA (forward primer TTTTCTGCTATGGGCAGCTC, reverse primer CTTGGACACGGTAGCAGAGG). These primers also contained the adapter sequences for the Nanopore PCR Barcoding Kit SQK-PBK004 as overhangs (forward primer overhang TTTCTGTTGGTGCTGATATTGC, and reverse primer overhang ACTTGCCTGTCGCTCTATCTTC). The Phusion High-Fidelity DNA Polymerase (New England Biolabs, Ipswich, MA, USA) was used for amplification. The reaction mixture had a total volume of 50 μL, containing 1× Phusion GC Buffer, 0.5 μM of each primer, 0.2 mM dNTPs, 0.02 U/μL Phusion High-Fidelity DNA Polymerase, and 100 ng of cDNA. The reaction conditions corresponded to 98 °C for 30 s; 35 cycles of 98 °C for 10 s, 63 °C for 30 s, and 72 °C for 5 min; and 72 °C for 10 min. Electrophoresis on 1% agarose gels was performed to verify PCR products.

### 2.5. Antisense Oligonucleotide Treatment

Minigene-transfected HEK293T cells were treated with AONs following guidelines and a protocol described previously [59]. Briefly, two antisense oligonucleotides (AONs) were designed to target the region surrounding the complex allele. One AON was designed to bind directly over one of the two variants making up the complex allele (NM_000350.2:c.1555-5882C>A), while the other was designed to bind over a cryptic donor site located in the proximity of the second variant making up the complex allele (NM_000350.2:c.1555-5784C>G). The sequences and characteristics of the AONs are listed in Table 3 (Section 3.4). AONs were purchased from Microsynth AG (Balgach, Switzerland) as 2′-MOE-PTO (2′-*O*-methoxyethyl-phosphorothioate) oligonucleotides. AONs were resuspended in Phosphate-Buffered Saline (PBS) to a concentration of 100 µM.

Transfection of reference and variant minigenes was carried out as described in Section 2.4. Delivery of AONs was performed 24 h after minigene transfections using Lipofectamine 3000 Transfection Reagent (Thermofisher Scientific, Waltham, MA, USA) with a final AON concentration of 0.5 µM, according to the manufacturer’s instructions. RNA extraction and cDNA synthesis were achieved as described in Section 2.4.

### 2.6. Nanopore Sequencing and Data Analysis

The minigene-specific transcripts’ PCRs (AON-treated and untreated) underwent nanopore sequencing and data analysis as previously described [60]. Briefly, purification of PCR products was performed with AMPure XP beads (Beckman Coulter Life Sciences, Indianapolis, IN, USA) using a 1:1.5 (PCR/beads) ratio according to the manufacturer’s instructions. Subsequently, PCRs were resuspended in 50 µL of 1× Tris-EDTA (TE) buffer (Integrated DNA Technologies, Coralville, IA, USA) and the concentrations were determined with the QuBit dsDNA High Sensitivity Assay kit (Thermofisher Scientific, Waltham, MA, USA). Dilutions at 10 ng/µL in ddH_2_O were prepared with a final volume of 24 µL.

The barcoded universal primers with rapid attachment chemistry from the Nanopore PCR Barcoding Kit SQK-PBK004 (Oxford Nanopore Technologies, Oxford, UK) were added to the PCR products with the indexing PCR by adding 25 µL of Long Amp Taq 2X Master Mix (New England Biolabs, Ipswich, MA, USA) and 1 µL of primers. The following conditions were used for the reactions: 94 °C for 1 min; 30 cycles of 94 °C for 30 s, 62 °C for 30 s, and 65 °C for 2 min; and 65 °C for 5 min. The PCRs were purified with AMPure XP beads using a 1:1 ratio and eluted in 22 µL of ddH_2_O. The concentrations were quantified with the QuBit dsDNA High Sensitivity Assay kit. The size distribution of indexing PCR products was verified with a Bioanalyzer High-Sensitivity DNA kit on a Bioanalyzer 2100 instrument (Agilent Technologies, Santa Clara, CA, USA). To finalise the sequencing library, purified indexing PCR products were pooled to a total of 75 fmol in 10 µL, and 1 µL of RAP was added to the pooled library to attach the rapid 1D sequencing adapters by incubating for 5 min at room temperature. The libraries were sequenced on an R9.4.1 (FLO-MIN106D) Flow Cell combined with a MinION Mk1C instrument (Oxford Nanopore Technologies, Oxford, UK) running the MinKNOW v.23.07.5 software, according to the manufacturer’s instructions.

The raw pod5 files were converted to FASTQ files with the wf-basecalling v.1.0.1 workflow on the EPI2ME v.5.1.3 platform (Oxford Nanopore Technologies, Oxford, UK). The FASTQ files were demultiplexed with the Barcoding Analysis tool on the MinKNOW v.23.07.5 software. Subsequently, read mapping was executed with minimap2 v2.26 with the “splice” option active [61]. The alignment files were sorted, indexed, and converted to the BAM format using samtools v.1.18 [62].

Transcript identification and quantification were performed using the script on Github (https://github.com/jordimaggi/Minigene_transcripts_quantification_Nanopore (accessed on 15 September 2024)), as previously described [60]. Unknown splice junctions were manually curated based on prediction software available on Alamut^®^ Visual Plus v.1.6.1 (Sophia Genetics, Rolle, Switzerland). If an unknown acceptor or donor site did not correspond to a predicted cryptic acceptor or donor site, the splice junction was considered incorrectly called and it was manually assigned to the most likely nearby acceptor or donor site.

### 2.7. Pseudoexon Detection in cDNA from Retinal Organoids

To verify the presence of pseudoexon-inclusive transcripts in WT retinal cells, we used cDNA extracted from 23-week-old (W23) retinal organoids derived from human induced pluripotent stem cells (hiPSCs), as previously described [63,64]. This timepoint was selected because it was characterised by a large population of mature photoreceptors expressing *ABCA4* (Appendix A). Primers binding to *ABCA4* exon 11 (forward primer ATGGCCAACTTCGACTGGAG), exon 12 (reverse primer TTCACGTGGGGTGGTAGAGA), and both pseudoexons identified in this study (ABCA4_pe11a TTCACGTGGGGTGGTAGAGA and ABCA4_pe11b TTCACGTGGGGTGGTAGAGA, both forward primers) were designed. Three different PCR reactions were tested on WT W23 retinal organoid cDNA: ex11 with ex12 primers, pe11a with ex12 primers, and pe11b with ex12 primers. PCR reaction mixes were composed of 1X HOT FIREPol Buffer B2 from the HOT FIREPol kit (Solis Biodyne, Tartu, Estonia), 0.5 mM MgCl_2_, 200 µM dNTPs, 0.2 µM of each primer, 0.025 U/µL HOT FIREPol DNA polymerase, and 20 ng of cDNA in a final volume of 20 µL. PCR reactions were run under the following conditions: 95 °C for 15 min; 35 cycles of 95 °C for 45 s, 60 °C for 1 min, and 72 °C for 1.5 min; and 72 °C for 10 min. Electrophoresis on 1% agarose gel was performed to verify PCR products. Additionally, pseudoexon-specific PCRs were run on a Bioanalyzer High Sensitivity DNA chip on a Bioanalyzer 2100 instrument (Agilent Technologies, Santa Clara, CA, USA) to verify the size of the main products. The concentration of all PCRs was quantified with the Qubit dsDNA High-Sensitivity Assay kit (ThermoFisher, Waltham, MA, USA).

### 2.8. Pseudoexon Detection in Bulk RNA-seq Data from Retinal Organoids

Bulk RNA-seq datasets from 7-week-old (W7), 14-week-old (W14), and W23 retinal organoids were collected from previous studies [63,64]. The raw sequencing data were merged and aligned to the human reference genome hg19 using STAR aligner v.2.7.10a [65]. The alignment output file was sorted, indexed, and converted to the BAM format with samtools v.1.18 [62]. The splice junction output (SJ.out) file was used to verify the presence of splice junctions overlapping with the pseudoexons detected in this study.

## 3. Results

### 3.1. Clinical Presentation

At the initial examination (age of 10 years), corrected visual acuity (VA) was 0.8 (partial) Snellen decimal in both eyes. Anterior eye examinations were normal. Fundoscopy revealed a grainy appearance at the fovea in both eyes, with the retinal vasculature and optic nerve heads both appearing normal. The SD-OCT (Figure 1) revealed subnormal foveal thickness (OD 83 µm and OS 89 µm) and loss of the photoreceptor layers distal to the internal limiting membrane. Imaging revealed horizontally oval areas of reduced FAF in the foveal and perifoveal area bordered by increased FAF (Figure 1). The ffERG was quantitatively and qualitatively normal in both eyes, consistent with a normal pan-retinal function but not excluding the possibility of small areas of localised dysfunction. The MF-ERG showed reduced traces centrally and paracentrally, slightly more so in the left eye, consistent with dysfunction of the cone system in the foveal and perifoveal area (Figure 1). The traces were horizontally and possibly also vertically slightly asymmetric in each eye, consistent with eccentric fixation.

VA gradually declined to 0.16 Snellen (right eye) and 0.10 Snellen (left eye) over 6 years. Foveal thickness declined in both eyes over time, from 83/89 µm initially to 53/57 µm after 6 years in the right and left eyes, respectively (Figure 1). At 4 years after the initial examination, the FAF images of both eyes began to exhibit a less homogeneous, more granular-appearing hypofluorescence, with decreased prominence of the hyperfluorescent borders and increased generalised hyperfluorescence over the macular region, as seen after 6 years in Figure 1. The MF-ERG appeared approximately stable over 6 years when assessed using the concentric ring analysis method, as seen in Figure 1. However, longitudinal interpretation was challenging due to a necessary change of the monitor used to present the stimuli after 4 years, and we cannot exclude unstable or inconsistent fixation between examinations.

### 3.2. Identification of a Candidate Pathogenic Complex Allele in ABCA4

WES variant analysis revealed three candidate heterozygous variants in the genes *ABCA4*, *ALMS1*, and *C1QTNF5* (Table 1). Since no second candidate variant was identified in *ABCA4* or *ALMS1*, both of which have been associated with recessive forms of retinal diseases, these variants alone could not explain the phenotype in the index patient. *C1QTNF5* has been associated with dominant IRDs; however, the variant (NM_015645.4:c.212C>A) was inherited from the unaffected mother.

Since the clinical phenotype resembled STGD1, LR-PCR was performed for the entire *ABCA4* locus for the index and the parents to search for a second candidate variant, as previously described [43]. This allowed for the identification of a complex allele composed of two deep-intronic variants in intron 11 (NM_000350.2:c.[1555-5882C>A;1555-5784C>G], inherited from the mother) *in trans* to the missense variant in exon 22 (NM_000350.2:c.3322C>T, p.(Arg1108Cys), inherited from the father) identified with WES (Table 1). The splicing predictions for this complex allele were mild (Appendix A). Subsequently, we performed WGS.

WGS analysis led to the identification of two additional candidate heterozygous deep-intronic variants in the genes *CFH* and *MFSD8* (Table 1). The *CFH* variant (NM_000186.3:c.3494-405A>G) is predicted to create a strong cryptic acceptor site (Appendix A) by the splicing prediction tools on Alamut Visual Plus v.1.6.1 (SSF, MaxEntScan, NNSPLICE, GeneSplicer). Conversely, the SpliceAI and Pangolin predictions for this variant are bland. The *MFSD8* variant (NM_152778.2:c.199-1334A>G) strengthens a cryptic donor site 5 bp upstream of the variant according to the prediction tools on Alamut Visual Plus, SpliceAI, and Pangolin (Appendix A). Nevertheless, both genes are associated with recessive diseases, and no second candidate variant was identified in either gene.

After WGS analysis, the *ABCA4* variants remained the most likely cause of the clinical phenotype. The clinical findings in the index patient were suggestive of STGD1, further supporting these candidate variants. The missense variant (NM_000350.2:c.3322C>T) has been identified in STGD1 patients several times [66,67,68,69,70]. In contrast, the deep-intronic complex allele has not been described in the literature, to the best of our knowledge. Figure 2A highlights that both variants that are part of the complex allele localise between a relatively weak cryptic acceptor site (average transformed Alamut scores of 33%) and a strong cryptic donor site (average transformed Alamut scores of 76%) at positions c.1555-5923 and c.1555-5750, respectively. SpliceAI and Pangolin predict the usage of these cryptic splice sites with low likelihood (Appendix A). The complex allele may affect exonic splicing enhancer (ESE) and silencer (ESS) binding site profiles over the region, according to computations from EX-SKIP, ESEfinder, and RESCUE-ESE (Figure 2B and Appendix A).

### 3.3. Deep-Intronic Complex Allele Substantially Increases “WT” Pseudoexon Inclusion

A minigene construct based on the pcDNA3_RHO_ex3-5_plasmid was constructed for the functional characterisation of the complex deep-intronic allele. A 6944 bp region of *ABCA4* intron 11 surrounding the complex allele (c.1554+4787−c.1555−2643) was cloned between *RHO* exons 3 and 5 (refer to Materials and Methods Section 2.4). The expected major (WT) transcript for this minigene construct would contain exclusively *RHO* exons 3 and 5, for a transcript length of 119 bp.

The expected WT transcript (T1, highlighted in green under the coverage plots in Figure 3) was detected by nanopore sequencing in both reference and variant minigene assays and represented 98.7% and 54.2% of the reads, respectively. The second most common transcript (T2, Figure 3 and Table 2) in the reference minigene assay included a pseudoexon corresponding to the region c.1554+6710-c.1554+6836 (referred to as *ABCA4* pseudoexon 11a). This transcript was identified in the reference and variant minigene assays in low abundance (0.74% and 0.04% of the reads, respectively). In contrast, the second most abundant transcript in the variant minigene (T3, Figure 3 and Table 2) was characterised by the incorporation of another pseudoexon (referred to as *ABCA4* pseudoexon 11b) that uses the cryptic acceptor and donor sites predicted by SpliceAI and Pangolin (c.1555-5923 and c.1555-5750, respectively). T3 was found in 44.6% of the variant minigene assay results, while it was represented by only 0.05% of the reads from the reference minigene. Additionally, a fourth low-abundance transcript (T4, Figure 3 and Table 2) that included both pseudoexons (*ABCA4* pseudoexon 11a and *ABCA4* pseudoexon 11b) was found for the variant minigene (0.4% of the reads).

These reference minigene results suggest that intron 11 of *ABCA4* contains at least two “wildtype” pseudoexons that are sporadically (<1% of the transcripts) spliced into the mature mRNA. Pseudoexon 11a is out-of-frame (p.(Cys519Leufs*42)) and pseudoexon 11b is in-frame, but leads to a premature stop gain (p.(Cys519_Asp2273delins22)). For these reasons, transcripts containing either (or both) pseudoexons are expected to undergo nonsense-mediated mRNA decay.

The variant minigene revealed that the deep intronic variants severely increase the likelihood that the *ABCA4* pseudoexon 11b is included in the mature mRNA (T3 and T4, 45.05% versus 0.05%). In this context, the WT transcript was reduced to 54.2% of all mature mRNA.

### 3.4. Antisense Oligonucleotide Rescue of Aberrant Splicing

To assess the possibility of rescuing the complex allele-induced aberrant splicing events, two antisense oligonucleotides were designed to block either the ESE motifs created by one of the variants (AON1; Figure 4 and Table 3) or the cryptic donor splice site (AON2; Figure 4 and Table 3).

The two AONs allowed for the correction of variant-induced aberrant splicing events with different efficiencies (Figure 7). AON1 partially restored correct splicing, increasing the WT transcript’s relative abundance from 54.2% in the untreated variant minigene (Figure 3 and Figure 7, Table 2) to 77.2% when AON1-treated (Figure 5 and Figure 7, Table 4). In the reference minigene, AON1 treatment had no effect on the transcripts’ relative abundance (Figure 5 and Figure 7, Table 4).

Similarly, AON2 treatment showed no adverse effects on the relative abundance of the transcripts for the reference minigene. However, AON2 led to an almost complete elimination of pseudoexon 11b from the transcripts in the variant minigene and, therefore, an almost complete rescue of aberrant splicing. In fact, transcripts T3 and T4 represented a total of 45.05% in untreated cells (Figure 3 and Figure 7, Table 2) and only 0.06% in AON2-treated cells (Figure 6 and Figure 7, Table 5). Simultaneously, AON2 treatment led to an increase in the correctly spliced transcript (T1), from 54.2% in the untreated cells to 99.2% in the treated cells (Figure 6 and Figure 7, Table 5).

### 3.5. Pseudoexon Detection in Retinal Organoids

Since pseudoexons 11a and 11b were detected in transcripts expressed from the reference minigene, we wondered whether these pseudoexons were included in transcripts under normal retinal physiological conditions. To verify this, we used cDNA derived from retinal organoids that were generated as described elsewhere [63,64]. PCRs with three different sets of primers that shared the same reverse primer (binding to exon 12 of *ABCA4*; Figure 8 and Table 6) were performed; the first set was intended to amplify the reference transcript and possibly those containing pseudoexons 11a and/or 11b (forward primer binding to exon 11 of *ABCA4*), the second set used a forward primer specific for pseudoexon 11a, and the third set used one specific for pseudoexon 11b.

The PCRs confirmed the presence of low-abundance transcripts that included pseudoexons 11a and 11b in the cDNA derived from retinal organoids differentiated from human “WT” induced pluripotent stem cells (Figure 8 and Table 6). Gel electrophoresis for the PCRs revealed a strong product corresponding to the expected size for the “reference” PCR and two very weak products for the pseudoexon-specific PCRs (Figure 8A). To further investigate the pseudoexon-specific PCRs, they were run on a Bioanalyzer High Sensitivity chip to verify the size of the main products (Figure 8B,C and Table 6), which confirmed the presence of both pseudoexons. Finally, the concentration of the PCRs was measured to estimate their relative abundance, which resulted in the reference transcript (*ABCA4* exons 11 to 12 splice junction) being ≈15X more abundant than those including either pseudoexon (Table 6).

To further corroborate this finding, RNA-seq data of retinal organoids from previous studies [63,64] was queried for evidence of splice junctions overlapping the pseudoexons identified in this study. The splice junction file generated by the STAR aligner listed 5 split reads connecting *ABCA4* exon 10 to pseudoexon 11a (maximum overhang of 73 bp), and 12 split reads connecting pseudoexon 11a to *ABCA4* exon 11 (maximum overhang of 60 bp). *ABCA4* exons 10 and 11 were connected by 993 split reads (maximum overhang of 90 bp), meaning that pseudoexon 11a was included in 0.5–1.2% of the transcripts (between 5/1005 and 12/1005 split reads spanning the region). Evidence of pseudoexon 11b inclusion could not be detected in the splice junction file.

## 4. Discussion

We report a novel pathogenic complex deep-intronic allele in *ABCA4* intron 11 (NM_000350.2:c.[1555-5882C>A;1555-5784C>G]) identified in a patient affected by childhood-onset Stargardt disease. The reference minigene assay (complex allele absent) indicated the existence of two low-abundance transcripts that included different ‘wildtype’ pseudoexons from *ABCA4* intron 11 (named pseudoexons 11a and 11b, respectively). Both pseudoexons could be detected in the cDNA of retinal organoids derived from induced pluripotent stem cells (iPSCs). Similarly, retinal organoid RNA-seq data provided evidence for the inclusion of pseudoexon 11a in a low-abundance transcript. The variant minigene assay revealed that the complex allele causes pseudoexon 11b to be incorporated in a substantially larger proportion of mature mRNA. Both variants form the complex allele overlap with pseudoexon 11b, but neither is predicted to affect its splice sites. Instead, the complex allele is predicted to modify the ESE/ESS ratio. Finally, we tested the ability of two AONs to rescue the aberrant splicing events detected in the variant minigene assay. The two AONs showed different efficiencies in modulating the complex allele-mediated aberrant splicing. AON2 was designed to block pseudoexon 11b’s donor splice site and resulted in complete rescue of correct splicing in the minigene assay.

Deep-intronic variants have been recognised as important contributors to disease in patients with STGD1 and other *ABCA4*-related IRDs [27,29,35,36,37,38,71,72,73,74,75]. The main challenge with deep-intronic variants in *ABCA4* is assessing their effect on splicing due to the unavailability of patient-derived retinal tissues. To overcome this challenge, some of these studies took advantage of mini- (and midi-) gene assays to functionally characterise their impact on splicing [29,37,38]. Other studies described the possibility of using alternative patient-derived cell types, namely fibroblasts and keratinocytes, to functionally test deep-intronic variants in *ABCA4* [35,72]. Finally, other studies used patient-derived iPSCs to generate differentiating photoreceptor cells, which were examined for variant-induced aberrant splicing events [74,75].

Each of the models mentioned above has advantages and disadvantages. Minigene assays are simplified gene models often restricted to short fragments of the investigated gene; this can facilitate result interpretation, but it may lead to oversimplified conclusions. We have previously shown that minigenes modelling the same variant with alternative inserts can result in the identification of different transcripts and their relative quantification may vary substantially [60]. Moreover, the choice of cell type to be transfected can influence aberrant splicing events in terms of their relative abundance and nature [76]. For these reasons, aberrant splicing events and their relative abundance detected with minigene assays may not exactly reflect the variant-induced events in the physiologically relevant cell type(s) or tissues. However, the detection of alternative splicing events when comparing the reference minigene to the variant minigene remains a strong indication that the variant can lead to aberrant splicing. Similarly, the use of alternative patient-derived cell types (i.e., keratinocytes and fibroblasts) can provide evidence of aberrant splicing events; however, cell type-specific splicing and the presence of other patient-specific sequence variants may confound results [77]. In addition, retinal organoids provide access to physiologically relevant cell types for the characterisation of candidate splicing variants in IRD-associated genes. The model, however, is time- and resource-consuming. Additionally, the presence of additional variants in patient-derived cells can confound the outcomes. Finally, hiPSC differentiation into retinal organoids can only provide immature and incomplete retinal tissue [78], which may influence splicing patterns.

The identification and characterisation of splicing-altering variants is critical, as it facilitates the development of AON-based therapies. We and others have demonstrated that design of personalised AON-based therapies to correct splicing patterns in vitro for *ABCA4* can be straightforward [28,29,39]. Preclinical and clinical studies for an AON-based rescue of splice defects caused by a relative common pathogenic deep-intronic variant in *CEP290* have demonstrated efficacy in restoring correct splicing and improving visual acuity in treated eyes [79,80,81,82]. AON-based therapies for pathogenic variants in exon 13 of *USH2A* and for a dominant variant in *RHO* are currently undergoing clinical trials [83,84]. Topical drops and intravitreal injections have shown penetration into the retina and favourable safety profiles [83].

As discussed above, the use of minigene assays to model splicing of retina-specific genes does not deliver conclusive evidence to classify variants as definitely pathogenic. Another limitation of this study is represented by the reliance on PCR amplifications, which may introduce important biases. Nevertheless, the current study emphasises that current splicing prediction software remains unable to reliably predict the effects of deep-intronic variants, in particular when these do not alter or create cryptic splice sites. Therefore, functional assays to verify the presence of variant-induced aberrant splicing events are required to interpret these kinds of variants. Moreover, minigene assays can identify wildtype pseudoexons that are sporadically included in mature mRNA, which represent regions at high risk of activation by non-splice site variants [85]. Finally, AON-based therapies hold great potential for STGD1 and other IRD patients.

## Figures and Tables

**Figure 1 genes-15-01503-f001:**
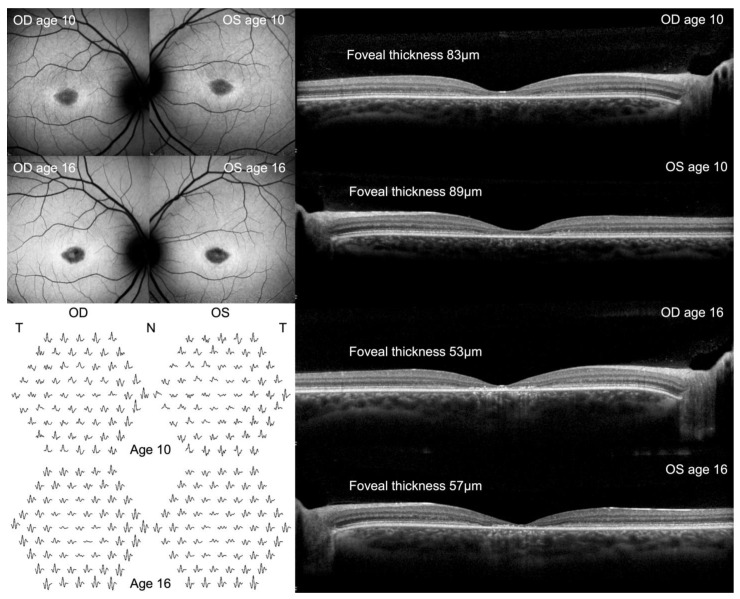
Fundus autofluorescence (FAF; **top left**), multifocal electroretinogram (MF-ERG; **bottom left**), and optical coherence tomography (OCT; **right**) findings from the index patient at 10 and 16 years of age. Worsening of the FAF and OCT structural findings was confirmed, however, the functional MF-ERG findings were approximately stable (although not directly comparable due to a change of monitor between examinations and likely eccentric fixation).

**Figure 2 genes-15-01503-f002:**
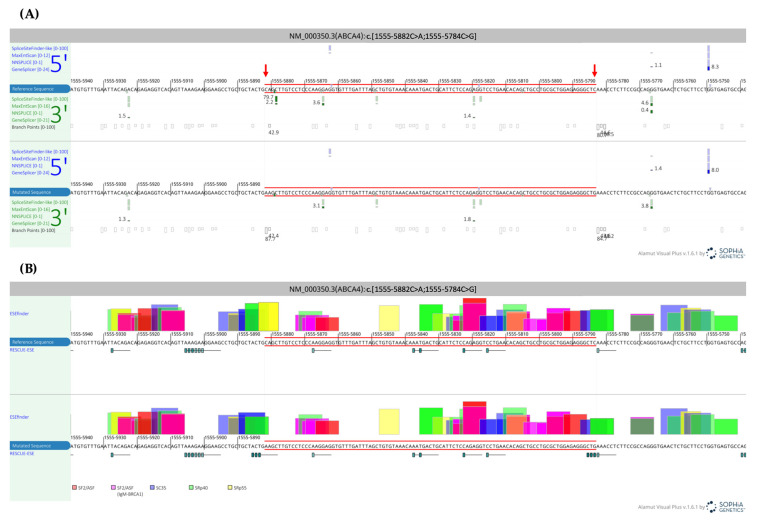
Visual representation of splicing prediction algorithms from the deep-intronic complex allele in *ABCA4*. The figure was created from two screenshots of the Alamut Visual Plus software v.1.6.1. (**A**) The panel shows the genomic region surrounding the complex allele NM_000350.2:c.[1555-5882C>A;1555-5784C>G] (reference sequence above the variant sequence), with the respective splice site predictions computed by the algorithms included in Alamut Visual Plus (SpliceSiteFinder-like, MaxEntScan, NNSPLICE, and GeneSplicer). Predicted acceptor and donor splice sites are represented by green and blue shapes, respectively. The arrowheads indicate the location of the two variants. (**B**) Screenshot showing exonic splicing enhancer (ESE) and silencer (ESS) binding region predictions from the ESEfinder v.3.0 and RESCUE-ESE tools.

**Figure 3 genes-15-01503-f003:**
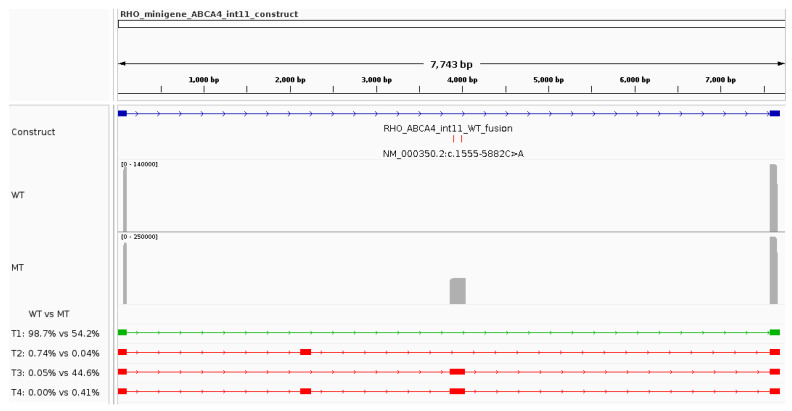
Functional characterisation of the *ABCA4* complex allele in intron 11 using a minigene assay. The IGV screenshot highlights the construct’s characteristics, the sequencing coverage plots for the reference (WT) and variant (MT) minigenes, and all transcripts (name T#) identified in the analysis. The relative abundance of each transcript in reference and variant minigenes can be seen underneath the coverage plots. The green transcript represents the expected major (WT) transcript.

**Figure 4 genes-15-01503-f004:**
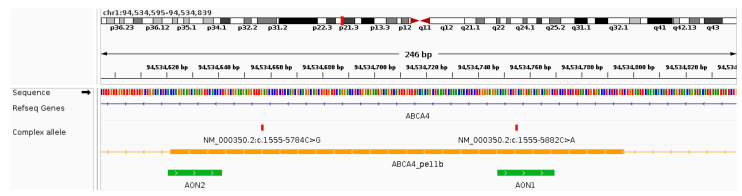
Antisense oligonucleotide (AON) binding sites relative to the complex allele and pseudoexon 11b. Screenshot from IGV over the genomic region chr1:94534595-94534839 (hg19) indicating the position of the variants that are part of the complex allele (red vertical lines), the position of pseudoexon 11b (orange horizontal bar), and the binding location of the antisense oligonucleotides tested in this study (AON1 and AON2; green horizontal bars). AON1 overlaps with variant c.1555-5882C>A and AON2 binds over the cryptic donor site.

**Figure 5 genes-15-01503-f005:**
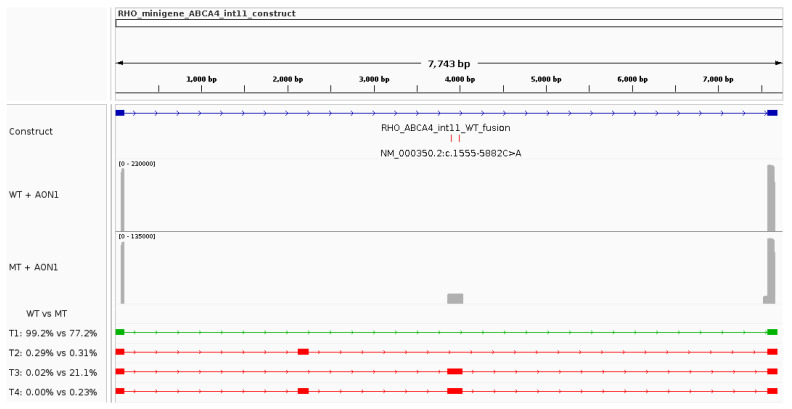
Aberrant splicing rescue assay for a deep-intronic complex allele in *ABCA4* with an antisense oligonucleotide targeting the first variant (AON1). The IGV screenshot highlights the construct’s characteristics, the sequencing coverage plots for the reference (WT) and variant (MT) minigenes treated with AON1, and all transcripts (name T#) identified in the analysis. The relative abundance of each transcript in reference and variant minigenes can be seen underneath the coverage plots. The green transcript represents the expected major (WT) transcript.

**Figure 6 genes-15-01503-f006:**
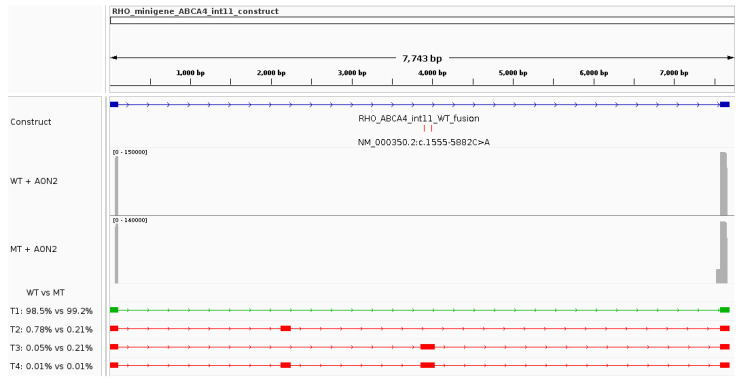
Aberrant splicing rescue assay for a deep-intronic complex allele in *ABCA4* with an antisense oligonucleotide targeting the cryptic donor site (AON2). The IGV screenshot highlights the construct’s characteristics, the sequencing coverage plots for the reference (WT) and variant (MT) minigenes treated with AON2, and all transcripts (name T#) identified in the analysis. The relative abundance of each transcript in reference and variant minigenes can be seen underneath the coverage plots. The green transcript represents the expected major (WT) transcript.

**Figure 7 genes-15-01503-f007:**
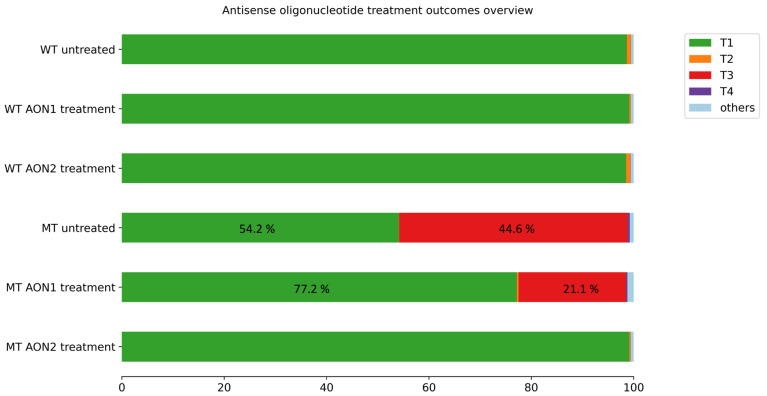
Antisense oligonucleotide treatment outcomes. Bar plot showing the relative abundance of all transcripts identified in the study (T1–T4) as well as that of unidentified transcripts (“others”) for both minigene sequences (reference minigene denoted as “WT” and variant minigene denoted as “MT”) in each experimental condition: untreated (results discussed in Section 3.3), treated with AON1, and treated with AON2. Abbreviations: WT, wildtype (or reference); MT, mutant (or variant); AON, antisense oligonucleotide.

**Figure 8 genes-15-01503-f008:**
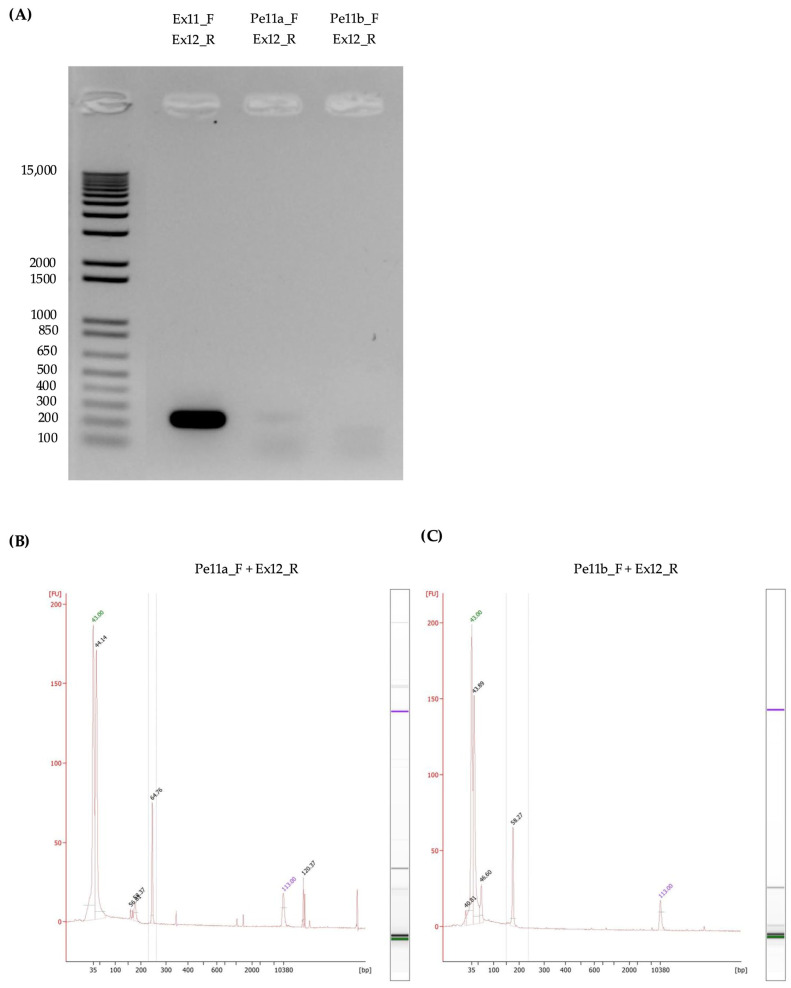
Pseudoexon 11a and 11b detection in cDNA derived from retinal organoids. (**A**) Gel electrophoresis results for the three PCRs to detect pseudoexons 11a and 11b within retinal organoid cDNA. Primer combinations are shown on top of the respective lane. The size in basepairs of ladder fragments is displayed on the left side. The image has been adapted for reasons of space by removing several irrelevant lanes; the original full-width gel image is available in the Appendix A). (**B**,**C**) Bioanalyzer traces from an Agilent DNA High Sensitivity chip for the pseudoexon-specific PCRs (Pe11a_F + Ex12_R and Pe11b_F + Ex12_R, respectively) highlighting the main PCR products (selected within vertical grey lines) and their sizes. The size in basepairs is displayed at the bottom. The intensity of the fluorescent signal is shown on the vertical axis in fluorescent units (FU).

**Table 1 genes-15-01503-t001:** Candidate variants detected in the WES, LR-PCR, and WGS datasets.

Gene	cNomen	gnomAD All (%)	ACMG	LOVD	ClinVar	HGMD	Ref.	Testing Assay
*ABCA4*	NM_000350.2:c.3322C>T	0.013	P/P	VUS/LP/P	LP/P	DM	[66]	WES/LR-PCR/WGS
*ABCA4*	NM_000350.2:c.1555-5784C>G	NA	VUS/LB	-	-	-	-	LR-PCR/WGS
*ABCA4*	NM_000350.2:c.1555-5882C>A	NA	VUS/LB	-	-	-	-	LR-PCR/WGS
*ALMS1*	NM_001378454.1:c.3177C>A	NA	LP/LP	-	-	-	-	WES/WGS
*C1QTNF5*	NM_015645.4:c.212C>A	0.004	VUS/VUS	VUS	VUS	-	-	WES/WGS
*CFH*	NM_000186.4:c.3494-405A>G	0.032	VUS/LB	-	-	-	-	WGS
*MFSD8*	NM_152778.3:c.199-1334A>G	0.298	VUS/LB	-	-	-	-	WGS

Abbreviations: cNomen, Human Genome Variation Society (HGVS) cDNA-level nucleotide change nomenclature; gnomAD all (%), genome aggregation database overall minor allele frequency in percentage; ACMG, American College of Medical Genetics and Genomics guidelines; LOVD, Leiden Open Variation Database; HGMD, Human Gene Mutation Database; Ref., reference; VUS, variant of unknown significance; P, pathogenic; LP, likely pathogenic; LB, likely benign; B, benign; DM, disease-causing mutation; NA, not available; WES, whole-exome sequencing; LR-PCR, long-range PCR sequencing; WGS, whole-genome sequencing.

**Table 2 genes-15-01503-t002:** Minigene transcript summary for the *ABCA4* complex allele NM_000350.2:c.[1555-5882C>A;1555-5784C>G]. The table provides an overview of the identified transcripts and their characteristics, such as length, their relative abundance in reference (WT) and variant (MT) minigenes, the difference (delta) in relative abundance between MT and WT sequencing results, the absolute number of reads representing each transcript, and the effect on the transcript. The table is sorted by relative abundance in the reference minigene.

	Transcript	Length	WT (%)	MT (%)	Δ MT-WT (%)	Counts WT	Counts MT	Effect on Transcript
T1	RHO_ex3-RHO_ex5	119 bp	98.69	54.15	−44.54	58,175	61,349	WT
T2	RHO_ex3-ABCA4_pe11a-RHO_ex5	246 bp	0.74	0.04	−0.7	438	49	pe11a
T3	RHO_ex3-ABCA4_pe11b-RHO_ex5	293 bp	0.05	44.64	44.59	30	50,572	pe11b
T4	RHO_ex3-ABCA4_pe11a-ABCA4_pe11b-RHO_ex5	420 bp	0	0.41	0.41	1	456	pe11a + pe11b

Abbreviations: WT, wildtype (or reference); MT, mutant (or variant); ex, exon; pe, pseudoexon; bp, basepairs; Δ, delta.

**Table 3 genes-15-01503-t003:** Antisense oligonucleotide (AON) sequences and their characteristics.

AON#	Sequence	Length	Tm	GC
AON1	ACAAGCTGCAGTAGCAGCAGG	21 bp	59.5	57.1
AON2	ACCAGGAAGCAGAGTTCACC	20 bp	56.0	55.0

Abbreviations: AON, antisense oligonucleotide; #, number; Tm, theoretical melting temperature in °C; GC, percentage of GC; bp, basepairs.

**Table 4 genes-15-01503-t004:** Minigene transcript summary for the *ABCA4* complex allele NM_000350.2:c.[1555-5882C>A;1555-5784C>G] for reference (WT) and variant (MT) minigenes treated with AON1. The table provides an overview of the identified transcripts and their characteristics, such as length, their relative abundance in reference (WT) and variant (MT) minigenes, the difference (delta) in relative abundance between MT and WT sequencing results, the absolute number of reads representing each transcript, and the effect on the transcript. The table is sorted by relative abundance in the reference minigene.

	Transcript	Length	WT (%)	MT (%)	Δ MT-WT (%)	Counts WT	Counts MT	Effect on Transcript
T1	RHO_ex3-RHO_ex5	119 bp	99.15	77.18	−21.97	89,966	36,962	WT
T2	RHO_ex3-ABCA4_pe11a-RHO_ex5	246 bp	0.29	0.31	0.02	262	150	pe11a
T3	RHO_ex3-ABCA4_pe11b-RHO_ex5	293 bp	0.02	21.05	21.03	26	10,082	pe11b
T4	RHO_ex3-ABCA4_pe11a-ABCA4_pe11b-RHO_ex5	420 bp	0	0.23	0.23	2	108	pe11a + pe11b

Abbreviations: WT, wildtype (or reference); MT, mutant (or variant); ex, exon; pe, pseudoexon; bp, basepairs; Δ, delta.

**Table 5 genes-15-01503-t005:** Minigene transcript summary for the *ABCA4* complex allele NM_000350.2:c.[1555-5882C>A;1555-5784C>G] for reference (WT) and variant (MT) minigenes treated with AON2. The table lists the transcripts identified, along with their characteristics, such as length, their relative abundance in reference (WT) and variant (MT) minigenes, the difference (delta) in relative abundance between MT and WT sequencing results, the absolute number of reads representing each transcript, and the effect on the transcript. The table is sorted by relative abundance in the reference minigene.

	Transcript	Length	WT (%)	MT (%)	Δ MT-WT (%)	Counts WT	Counts MT	Effect on Transcript
T1	RHO_ex3-RHO_ex5	119 bp	98.53	99.15	0.62	58,968	44,223	WT
T2	RHO_ex3-ABCA4_pe11a-RHO_ex5	246 bp	0.78	0.21	−0.57	468	94	pe11a
T3	RHO_ex3-ABCA4_pe11b-RHO_ex5	293 bp	0.05	0.05	0	33	24	pe11b
T4	RHO_ex3-ABCA4_pe11a-ABCA4_pe11b-RHO_ex5	420 bp	0.01	0.01	0	4	4	pe11a + pe11b

Abbreviations: WT, wildtype (or reference); MT, mutant (or variant); ex, exon; pe, pseudoexon; bp, basepairs; Δ, delta.

**Table 6 genes-15-01503-t006:** Overview of PCRs for the detection of pseudoexons 11a and 11b from cDNA derived from retinal organoids, including the expected length of the main product, the size calculated by the Bioanalyzer software for the pseudoexon-specific PCRs, and the concentration measured with Qubit High Sensitivity dsDNA.

PCR Primers	Expected Length	Measured Length	Conc. (ng/µl)
Ex11_F + Ex12_R	224 bp	-	47
Pe11a_F + Ex12_R	239 bp	245 bp	3.2
Pe11b_F + Ex12_R	171 bp	174 bp	2.1

Abbreviations: ex, exon; pe, pseudoexon; bp, basepairs; ng, nanogram; µl, microlitre.

## Data Availability

The original contributions presented in the study are included in the article/Appendix A; further inquiries can be directed to the corresponding author/s.

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
