# Peer review of "Rescue of Aberrant Splicing Caused by a Novel Complex Deep-intronic ABCA4 Allele"

_genes, 2024, doi:10.3390/genes15121503_

Round 1
Reviewer 1 Report
Comments and Suggestions for Authors
The manuscript describes a case of an early-onset STGD1, due to a possible pathogenic variant in ABCA4 gene, causing an aberrant splicing site. This manuscript describes new aberrant splicing events due by a novel deep-intronic complex allele in intron 11 of ABCA4 identified in a STGD1 patient in trans to a pathogenic missense variant. This report is a relevant subject in the field since it can contribute to a better understanding of STGD1.
The overall manuscript is well written and it has impact in identifying new variants causing IRDs.
However, one issue must be addressed:
It seams that in Figure 8 A) the lane from the MWM does not corresponds to that gel. It is possible to see that the image was, somehow manipulated. Please, present the raw data regarding this image.
Reviewer 2 Report
Comments and Suggestions for Authors
Dear Authors,
First of all, congratulations for your interesting work. I hope that my hints will help you in the next steps of improvement and the final manuscript will be really valuable for the readers. Indeed, the topic you've tackled is rare and very important. Congratulations for your determination in creating this elegant model of the disease, time-consuming and effort-full process.
Your paper is very technical and every step of the work has been precisely described. This is very good in the main body of the paper, however I would recommend some tiny changes in the abstract - it should be more "popular-science" instead of technical, should explain potential readers why they really should read entire work, be catchy and interesting. I have the impression I'm reading "materials and methods" section, which shouldn't be the case.
There are several punctation mistakes (such as double space, double dot or no at all) and some typos , as well as the same word used sometimes with first capital letter, and next only small letters etc. - even if they do not change the value of the manuscript, I'd like to urge you to correct these imperfections.
It might be a good idea to underline the importance of intronic (and generally located in non-coding regions) variants in diagnostics, clinics; hence, the importance of analysing more than one gene, more than selected mutations. Unfortunately, in many countries "genetic testing" means analysing only a couple of variants, not even one gene.
Not every abbreviation used has been explained, correct please. It might be a good idea to add a list of abbreviations used.
Line 80 --> why only 50-75% of patients? is it because rarely WGS is performed? Or maybe this is because of the interpretation?
Finally, ithe paper is very dense and might be hard to digest, especially for younger readers. Could you think about adding some pictures, schemes? This applies even to the introduction, which is nicely described.
Nevertheless, it's a very good paper and I'm happy I had an opportunity to read it, congratulations.
P.S. Do not waste time to answear me here, just work on your excellent paper. thanks! :)
Round 2
Reviewer 1 Report
Comments and Suggestions for Authors
Thank you very much for addressing the issues mentioned before.